# Downscaling livestock census data using multivariate predictive models: Sensitivity to modifiable areal unit problem

Daniele Da Re[1]*, Marius Gilbert[2], Celia Chaiban[1,2], Pierre Bourguignon[1], Weerapong Thanapongtharm[3], Timothy P. Robinson[4,5], Sophie O. Vanwambeke[1]

**1** George Lemaitre Centre for Earth and Climate Research, Earth and Life Institute, UCLouvain, Louvain-la-Neuve, Belgium, **2** Spatial Epidemiology Lab (SpELL), Université Libre de Bruxelles, Brussels, Belgium, **3** Department of Livestock Development (DLD), Bangkok, Thailand, **4** Policies, Institutions and Livelihoods (PIL), International Livestock Research Institute (ILRI), Nairobi, Kenya, **5** Livestock Information, Sector Analysis and Policy Branch (AGAL), Food and Agriculture Organisation of the United Nations (FAO), Rome, Italy

* daniele.dare@uclouvain.be

**Data Availability Statement:** The R codes used in the study and the aggregated census data at different administrative levels are available in the

## Abstract

The analysis of census data aggregated by administrative units introduces a statistical bias known as the modifiable areal unit problem (MAUP). Previous researches have mostly assessed the effect of MAUP on upscaling models. The present study contributes to clarify the effects of MAUP on the downscaling methodologies, highlighting how a priori choices of scales and shapes could influence the results. We aggregated chicken and duck fine-resolution census in Thailand, using three administrative census levels in regular and irregular shapes. We then disaggregated the data within the Gridded Livestock of the World analytical framework, sampling predictors in two different ways. A sensitivity analysis on Pearson's *r* correlation statistics and RMSE was carried out to understand how size and shapes of the response variables affect the goodness-of-fit and downscaling performances. We showed that scale, rather than shapes and sampling methods, affected downscaling precision, suggesting that training the model using the finest administrative level available is preferable. Moreover, datasets showing non-homogeneous distribution but instead spatial clustering seemed less affected by MAUP, yielding higher Pearson's *r* values and lower RMSE compared to a more spatially homogenous dataset. Implementing aggregation sensitivity analysis in spatial studies could help to interpret complex results and disseminate robust products.

## Introduction

Spatial data are becoming increasingly more accessible to the scientific community. However, much data are provided in an aggregated form at different administrative levels, mainly for operational and privacy reasons [1, 2]. Administrative levels are usually determined and modifiable, meaning that they can be subdivided to form units of different sizes and shapes [3, 4].

gitlab folder https://gitlab.com/danidr/glw/tree/master/glw_maup.

**Funding:** D.D.R. is supported by the FRFS-WISD Walloon Institute for Sustainable Development PDR "Mapping livestock's transition" (PDR-WISD X302317F). The funders had no role in study design, data collection and analysis, decision to publish, or preparation of the manuscript.

**Competing interests:** The authors have declared that no competing interests exist.

Because administrative units may not adequately reflect the spatial organization of human or natural phenomena, researchers pursue the elaboration of methods for data disaggregation with the help of broadly available remote sensing data. Often, little attention is paid to the issue of the modifiable units and its effect on spatial representations [5]. This specific issue has been discussed in the spatial analysis literature since the 1930s (e.g. [6]), but gained attention with the milestone work of Openshaw and Taylor [7, 8] that led to the introduction of the concept of Modifiable Areal Unit Problem (MAUP). The MAUP encompasses two related but distinctive components: the scale issue and the zonation issue [3, 4, 7–10]. The scale problem reflects how the description of a phenomenon is potentially affected by changing the size of the sampling units, while the zonation issue relates to how changing the shape of sampling units could influence the representation of the phenomenon [7]. These effects occur because patterns and processes operate in the real world according to various scales and designs that are often unknown to the researcher [9]. A descriptive example illustrates some immediate effects. Fig 1a shows how the aggregation of individual-level data at different scales causes a reduction of the variability, and thus narrowing of the distribution. In Fig 1b, individual-level data are aggregated at the same scale but using different, arbitrary, areal unit shapes. The results are highly variable [3, 8, 10].

MAUP is closely related to the ecological inference fallacy, a misinterpretation of statistical inferences drawn at the group level but interpreted at the individual level [11]. With spatial data becoming a staple in a diversity of fields, the effects of MAUP have been widely explored, from ecology to remote sensing and from physical geography to economy [3, 10, 12–18]. Despite the fact that the impact of MAUP is often ingnored [5], when it is addressed researchers mostly assess its effect on upscaling, or aggregating [3, 16, 18], and mostly on its effect on model estimates rather than on downscaling, or disaggregating precision (but see [19]).

The availability of spatial data and data processing capacity fostered an interest into the spatial heterogeneity of diverse processes and encouraged researchers to find ways to disaggregate data. Downscaling techniques are used to disaggregate variables recorded or distributed at an aggregated scale, such as census data, and provide predictions at a finer level of spatial detail. Such fine scale data are of crucial interest in diverse fields and applications in agricultural socio-economics, food security, environmental impact assessment and epidemiology [20]. Concerning livestock, analyzing the emergence of zoonotic diseases requires detailed spatially explicit data of both hosts and their pathogens, e.g. for pathogenic avian influenza (HPAI, [21]).

The Gridded Livestock of the World (GLW, [22]) and WorldPop [23] disaggregate population data using statistical techniques and environmental predictors. Outputs of both projects attain good accuracy scores [20, 24], but as they result from a downscaling process, both are potentially subject to the MAUP. Despite the fact that the application of the GLW methodology has become robust and its application frequent (e.g. [20, 25–28]), its vulnerability to MAUP has not yet been directly investigated. Previous studies (e.g. [25]) showed a certain degree of sensitivity to the scale issue, however, the severity of the problem has not been assessed and a sensitivity analysis using various scale and shape configurations would help quantifying potential sources of uncertainties.

In this study, we analyzed the impact of both MAUP effects on the disaggregation of census-like livestock data. The objectives were: (i) to assess, on two different spatially-constrained real datasets, how the MAUP affects both goodness-of-fit metrics and downscaled results, (ii) to increase awareness about the MAUP issues in the context of data disaggregation. A fine resolution census dataset of poultry in Thailand was aggregated at scales corresponding to administrative levels, using sampling units with variable shapes and areas and subsequentely disaggregated to a common resolution over a 500m grid.

## a) Effects of aggregation

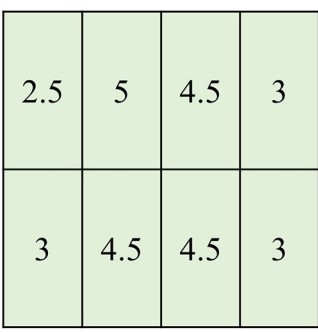

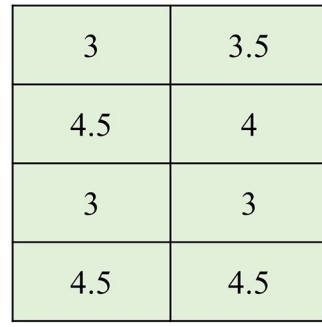

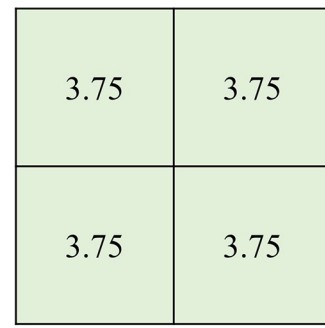

| | |
|---|---|
| Mean = 3.75 | |
| Std = 2.60 | |

Mean = 3.75
Std = 0.50

Mean = 3.75
Std = 0.00

## b) Effects of zoning systems

Mean = 3.75
Std = 0.93

Mean = 3.75
Std = 1.04

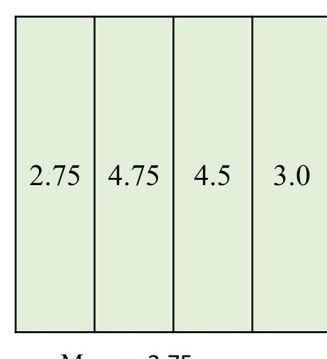

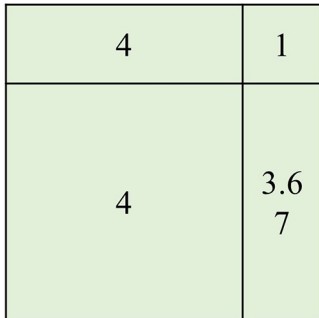

Mean = 3.17
Std = 2.11

**Fig 1. The modifiable areal unit problem.** Example showing the two effects of MAUP (adapted from [3]).

## Materials and methods

### Poultry population data

In 2010, the Department of Livestock Development of the Thai government conducted a national census of poultry in each sub-district and village, counting poultry head per owner. Each farm was associated by a unique administrative code number to its village, for which geographic coordinates were recorded. The census distinguished between broiler chickens, layer chickens, native chickens, farm ducks and free-grazing ducks. Here, we combined all data to species level ending with chicken and duck. The spatial constraints and determinants of the production systems of duck and chickens differ (intensive and backyard; [29–31]). While chickens can be raised anywhere, in Thailand, ducks are largely raised in wetlands used for double-crop rice production, where free-grazing ducks feed year round in rice paddies [30, 31].

Village records with incorrect coordinates (coordinates outside of the Thai territory or with 0 in latitude or longitude fields) were removed. In the case of duplicate coordinates or duplicate village unique ID, only one record for each duplicate was randomly selected. The provinces of Bangkok, Nakhon Sawan, Pattani and Phetchaburi were excluded due to lack of data. Once filtered, the village dataset was joined to the census dataset using the villages' administrative code number.

The poultry census individual level data were aggregated using a simple additive aggregation method according to Thai administrative units: districts, sub-districts and villages. As a comprehensive file of village boundaries is not available, Voronoi polygons were computed from the village coordinates.

## Modelling

We used the methodology of the Gridded Livestock of the World (GLW) project. The GLW disaggregates livestock statistics and provides spatially detailed estimates of livestock density in the form of raster spatial data [22]. The most recent version (GLW3; [26]) relies on stratified random forest models and a set of environmental predictors. The GLW methodology is fully described in [25] and [20]. Two user-controlled parameters drive the performance of random forest models: the number of trees created and the number of variables randomly selected when creating a splitting point. [32] have shown that 500 trees are a good rule of thumb, while the minimum number of variables that are randomly selected was calculated using the square root of the total number of variables [33].

The set of predictors was chosen among those shown to be relevant environmental and socio-economical drivers of poultry distribution [20, 30, 31, 34]. It included Fourier-transformed MODIS variables (two vegetation indices, the day and night land surface temperature and the band 3 middle-infra-red), eco-climatic variables (length of the growing season and annual precipitation), topographic variables (elevation and slope) land cover classes and anthropogenic variables (human population density and travel time to major cities and ports). Unpopulated areas, natural areas and water bodies were masked out and only areas suitable for poultry production were considered and used to get corrected poultry densities. Poultry densities corrected by area were transformed to logarithm (base 10) and used as response variable. The full list of spatial domain and predictors is detailed in Table 1 along with sources.

All input raster layers (e.g. masks and predictor variables) and outputs (predicted densities) were processed on the whole of Thailand with a spatial resolution of 500 m.

**Table 1. List of input spatial dataset used to model chickens and ducks densities.**

| Type | Variables | Use | Source |
|---|---|---|---|
| *Land* | Land and water area | Spatial domain and Spatial predictor | [35, 36] |
| *Land use* | IUCN world database of protected area | Mask | [37] |
| *Anthropogenic* | Worldpop human population density | Spatial predictor and suitability mask | [23] |
| | Travel time to the capital, province capitals and main harbors | Spatial predictor | [38, 39] |
| *Topography* | Elevation (GTOPO30) | Spatial predictor | [40] |
| | Slope (GTOPO30) | Spatial predictor | [40] |
| *Vegetation* | 10 Fourier-derived variables from Normalized Difference Vegetation Index from MODIS (MODIS)* | Spatial predictor | [41] |
| | Length of growing period | Spatial predictor | [42] |
| | Green-up and senescence (annual cycle 1 and 2) | Spatial predictor | [43] |
| | Forest cover | Spatial predictor | [44] |
| | Cropland, irrigated cropland and rainfed cropland cover | Spatial predictor | [45] |
| *Climatic* | 10 Fourier-derived variables from Day/Night Land Surface Temperature (MODIS) | Spatial predictor | [41] |
| | Precipitations | Spatial predictor | [46] |

*Annual mean, annual muinimum, annual maximum, amplitude and phase of annual cycle, amplitude and phase of bi-annual cycle, amplitude and phase of tri-annual cycle, variance in annual, bi-annual, and tri-annual cycles.

## Experimental design

The effect of scale was explored by aggregating the individual level data to village, sub-district and district level. The effects of zoning were analyzed using two different sets of polygon sampling units (PSUs) for each administrative level: (i) irregular (IRR) shapes, the original administrative units, and (ii) regular shapes (REG), a grid having the spatial resolution of the average spatial resolution (ASR) of the correspondent IRR PSUs. The ASR measures the effective resolution of administrative units in kilometers. It is calculated as the square root of the land area of the administrative units considered, divided by the number of administrative units [47, 48]. District, sub-district and village ASR is respectively 557.04, 69.55 and 8.30 km. REG PSUs were computed only at sub-district and district level. The density of birds per $km^2$ of suitable land was estimated in all polygons corresponding to each PSUs and transformed to its Log10 [25].

Two methods were applied to extract or sample the predictors by polygon, in order to understand their effect on the downscaled prediction. One method randomly sampled a point in each PSU and extracted the matching pixel value for each predictor. The other averaged the predictors within the PSU.

## Model evaluation

The polygons used as response variable were separated in training and validation sets. 70% of polygons were used to train the model, while the remaining 30% were used as evaluation data set. PSUs were sampled into training and evaluation datasets 20 times to assess the internal variability of the predictions. Once the model was fitted, average and standard deviation maps were computed from the 20 outputs.

Model evaluation was carried out using two approaches. Firstly, to assess how well the model predicted poultry densities, the root mean square error (RMSE) and Pearson's *r* correlation coefficient (COR) were computed between the observed values of the evaluation set of PSU and the predicted densities aggregated at polygon level of the corresponding validation PSUs. RMSE measures model accuracy, i.e. how far the predicted values were, on average, from the observed values. COR measures precision, i.e. the extent to which the observed and predicted values are proportional to each other. Lower RMSE and higher COR indicate better fits between predicted and observed values. RMSE and COR were estimated for the overall models. Moreover, to measure the internal precision associated with the area, RMSE and COR were also estimated considering PSUs area, grouping PSUs according to the frequencies of their area (Supporting information, S1 Fig): 0-10 $km^2$, 10-20 $km^2$ and >20 $km^2$ for villages, 0-100 $km^2$, 100-200 $km^2$ and >200 $km^2$ for sub-districts, 0-500 $km^2$, 500-1000 $km^2$ and >1000 $km^2$ for districts.

Secondly, Pearson's *r* was computed between predictions and the observed data at the village level only to assess the capacity of models trained using various PSUs to predict poultry population at a fine scale, i.e their "downscaling precision" ($COR_{down}$). This is crucial to understand the effects of MAUP on the downscaled predictions considering the finest administrative levels available as reference. Three different bounding boxes (hereafter *bbox*) were selected in different areas of Thailand to visually investigate the differences between the predictions and the observations. A graphical summary of the methodology is shown in Fig 2. The model is fully operational under R 3.4 [49] and the codes used, as well as the aggregated census data, are available at https://gitlab.com/danidr/glw/tree/master/glw_maup.

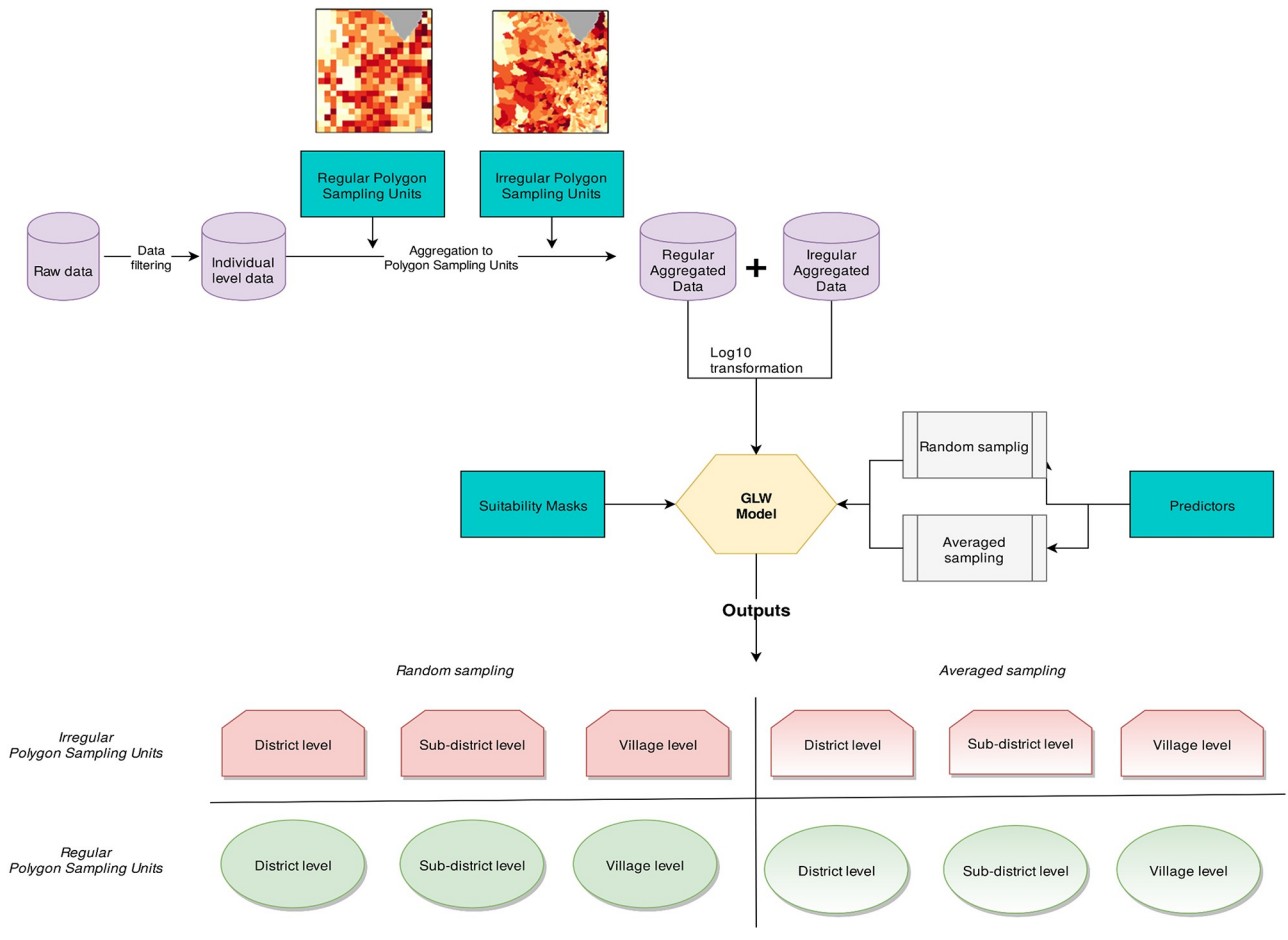

**Fig 2. Flowchart of the analysis.**

## Results

### Data cleaning and filtering

The 62 142 village records originally available were reduced to 57 794 (Table 2). Once the filtered village database was joined to the poultry census, the final georeferenced census dataset used to train the models accounts for 53 301 records (Table 2). Fig 3 show the observed densities for the two species aggregated at sub-districts and districts administrative level. Chickens were homogenously distributed. Ducks were mainly clustered in the central and southeast part of the country.

### Model output maps

The model predictions within *bbox* 1 are shown in Fig 4, while *bbox* 2 and 3 are displayed in the S4 and S7 Figs. Chickens were widely distributed though high density clusters are

**Table 2. Data filtering results.** For duplicate coordinates or duplicate village unique ID, only one record for each duplicated row was randomly selected and added to the finale database.

|  | Unfiltered | Duplicated ID | Duplicated coordinates | Filtered |
|---|---|---|---|---|
| *Villages* | 62142 | 6579 | 33 | 57794 |
| *Census* | 3170213 | - | - | 53301 |

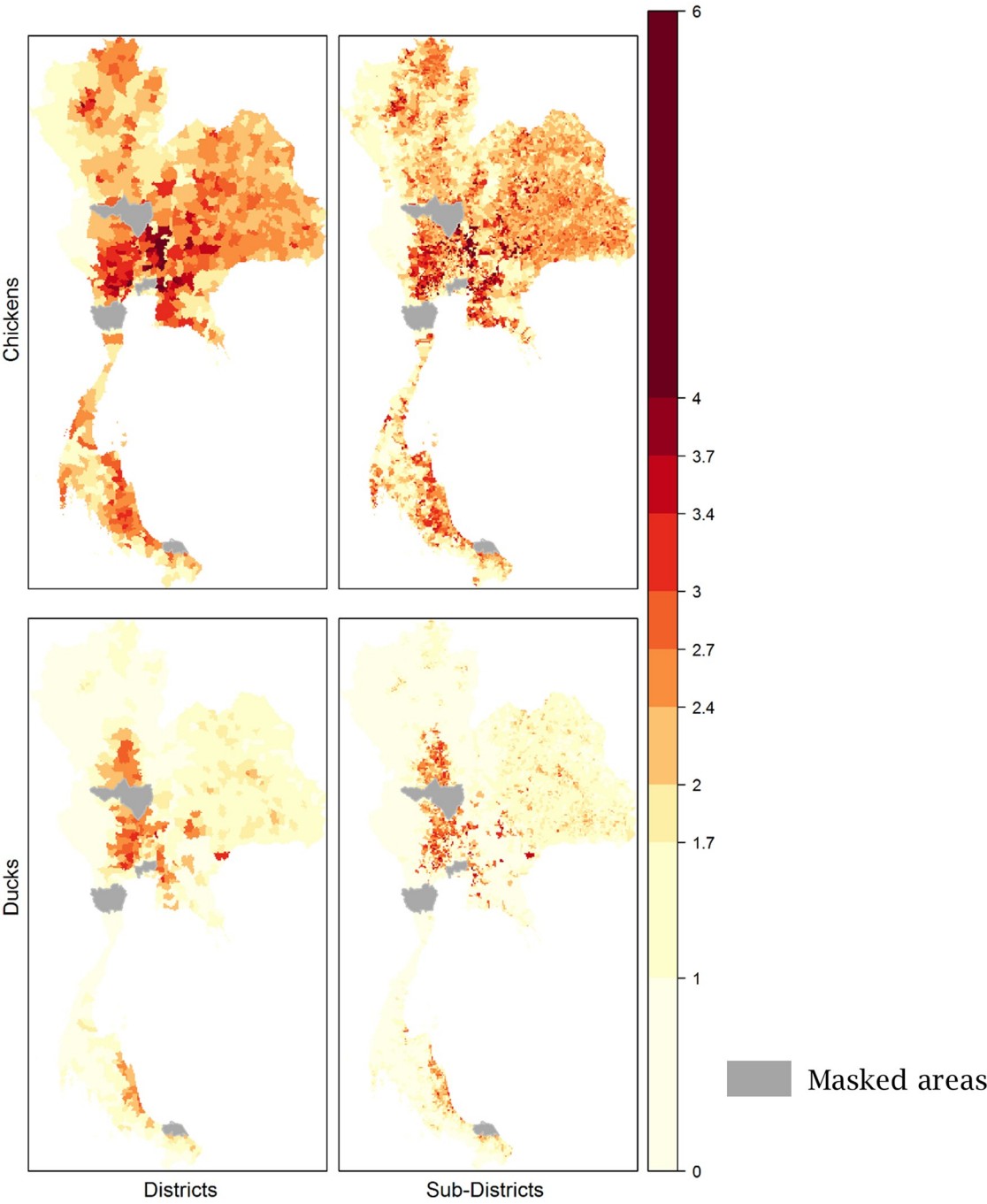

**Fig 3. Observed poultry densities in logarithm (base 10) aggregated at districts and sub-districts level.** In grey the provinces of Bangkok, Nakhon Sawan, Pattani and Phetchaburi, excluded from the analysis due to lack of data.

observable in the North-East and South-West parts of *bbox* 1. Ducks were present mostly in the central part. The model was able to reproduce the observed spatial pattern of both species, regardless of the sampling method.

The mean predicted values are comparable to the observed ones but the predicted values distributions are clustered around the mean and appeared less variable than the observed. For

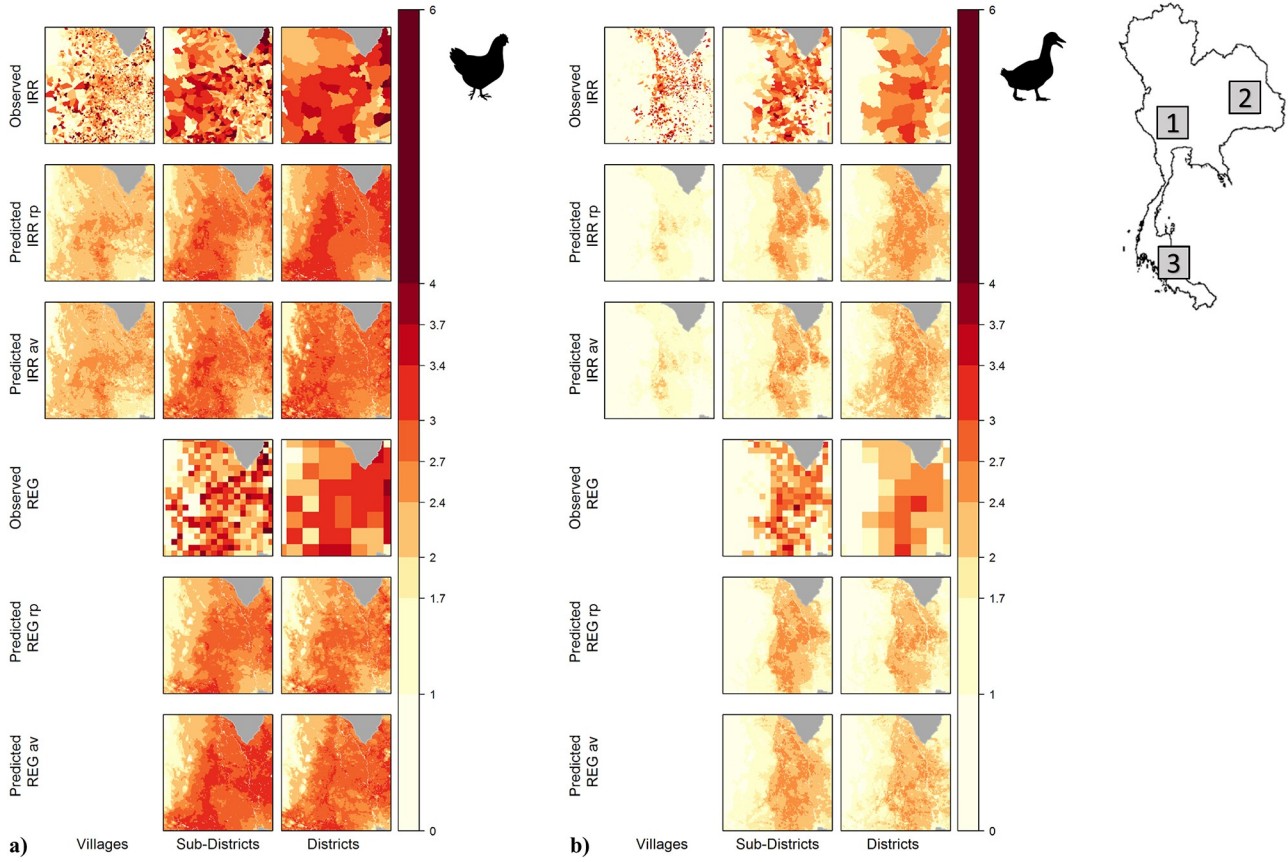

**Fig 4. Observed and predicted Log10 poultry values inside *bbox*1.** a) chickens, b) ducks.

both species, the aggregation of input data produced higher mean values at coarser scale, together with a narrowing effect of the value distribution and a smoothing effect on the frequencies (S2 and S3 Figs).

IRR and REG shaped administrative units showed slightly different predicted spatial patterns. In both cases, the distribution of the predicted values is consistent with the observed values, however, REG shapes seemed to predict a slightly smoother spatial pattern, detecting more variability across space than IRR shapes, which predicted more values clustered around and above the mean value.

## Model evaluation

The RMSE bar plots for ducks and chickens are shown in Fig 5. For both species, the overall accuracy increased (lower RMSE values) as the administrative level of the input data became coarser. However, this trend is more consistent for ducks rather than for chickens. Model runs on REG shaped PSUs showed generally less variability, but they had lower accuracy than IRR PSUs for chickens and comparable or slightly lower for ducks. Randomly sampling the predictors within the PSUs yielded slightly lower RMSEs than their aggregation.

COR bar plots based on stratified random sampling of the predictors and averaged predictors are shown in Fig 6. For both species, the COR value increased as the administrative level of the input data became coarser. REG PSUs produced higher correlations than the corresponding IRR PSUs and the overall models, showing also less variability among the bootstraps.

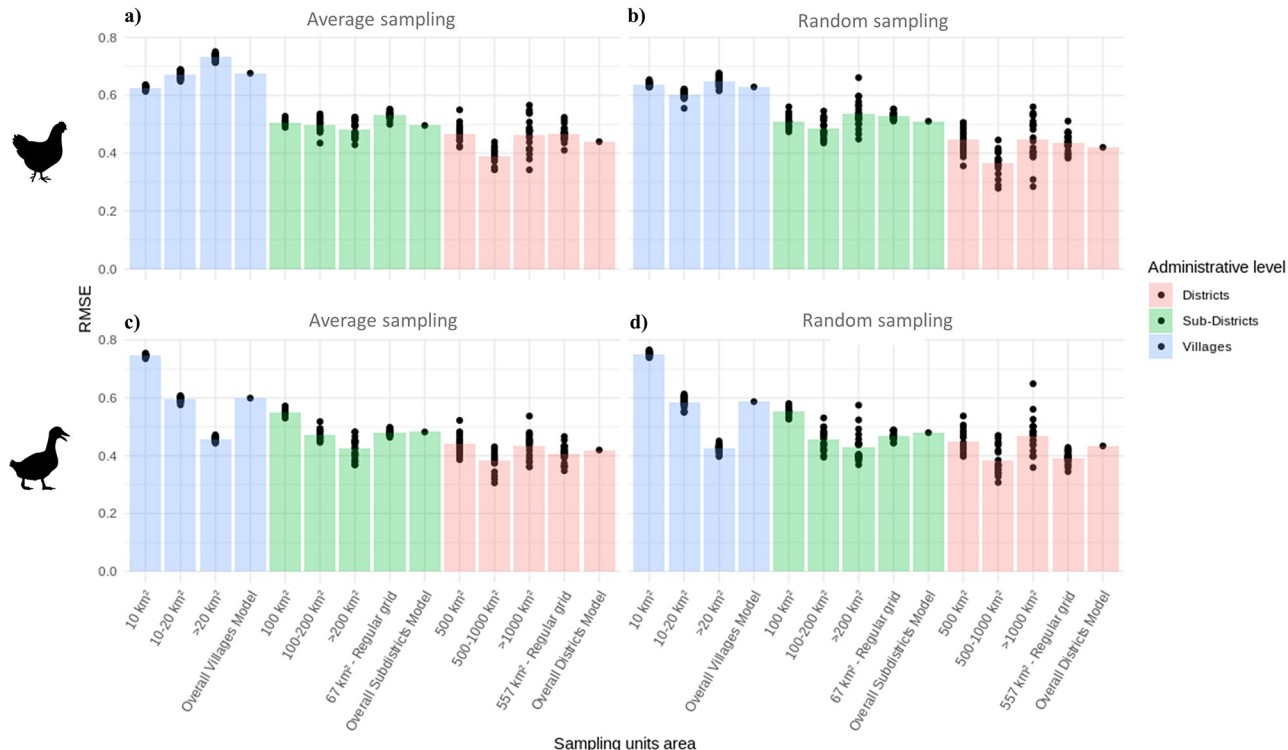

**Fig 5. Root mean square error (RMSE).** RMSE computed between predicted densities and observed chickens densities a) averaged sampling b) random sampling; RMSE computed between predicted densities and observed ducks densities c) averaged sampling d) random sampling.

The choice of the sampling methods did not affect the results strongly, but random sampling showed apparently higher variability between individual bootstraps.

## Downscaling precision

COR$_{down}$, the Pearson's $r$ coefficient between the predicted and observed densities at village level are shown in Fig 7. Models of duck distribution had higher correlations than the chicken models. Contrary to the internal precision of the model, smaller PSUs had higher Pearson's $r$ values than larger ones. The shape of the PSUs produced comparable results in terms of Pearson's $r$ values. Random sampling produced higher Pearson's $r$ values compared to average sampling, which generally had a lower variability among model runs. A table summarising the evaluation of model runs is found in S1 Table.

## Discussion

### Overall MAUP bias

Our model predicted poultry density patterns and value distributions similar to the observed densities, confirming the validity of the methodology [20]. As expected, chickens were dispersed at high densities across the whole country, while ducks were constrained to wetlands used for double-crop rice production [21, 30, 31].

The scale of the training data affected the output maps goodness-of-fit. On average, duck models showed higher downscaling precision and higher accuracy and precision compared to chickens. Swift, Liu and Uber [50] and Swift et al. [14] reported that a spatially clustered phenomenon aggregated using various size and shapes of areal units is less affected by MAUP

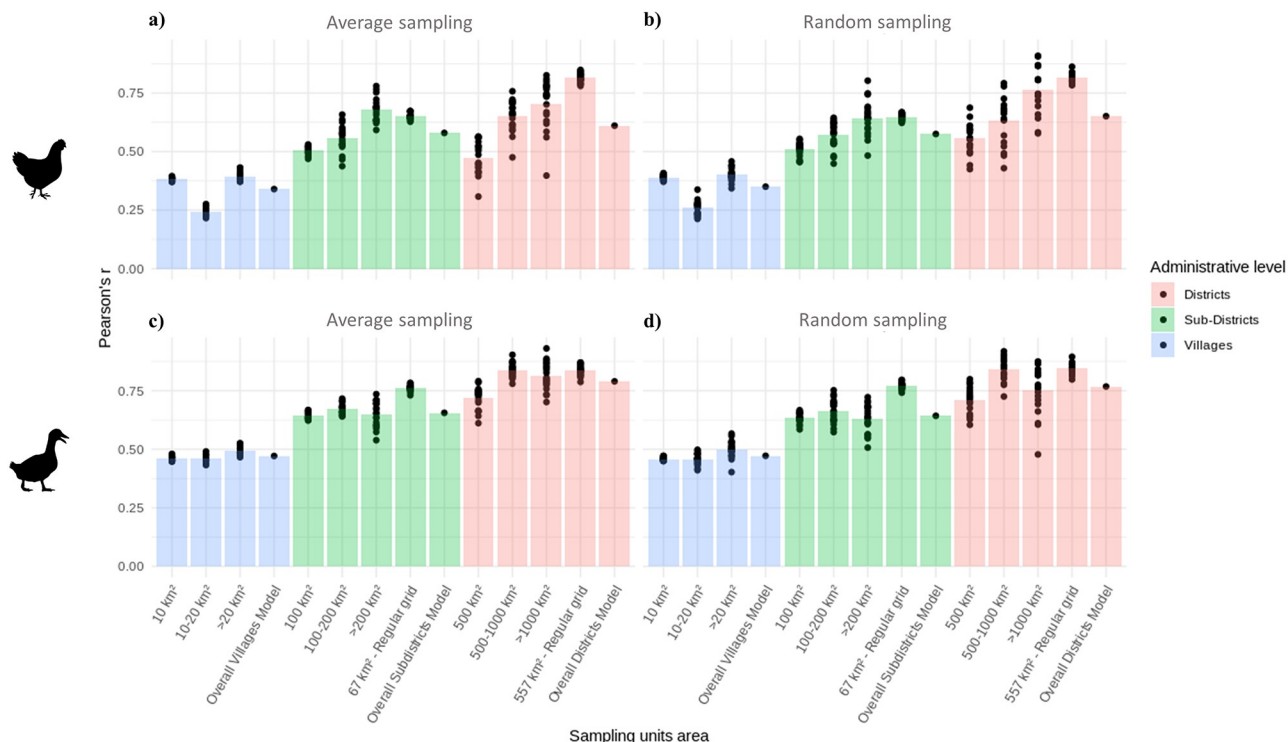

**Fig 6. Pearson's r.** Pearson's *r* coefficient computed between predicted densities and observed chickens densities a) averaged sampling b) random sampling; Pearson's *r* coefficient computed between predicted densities and observed ducks densities c) averaged sampling d) random sampling.

compared to a randomly distributed phenomenon. Because of that, when the clustered structure of the observed point pattern is preserved, the MAUP bias is considerably reduced. Moreover, Swift et al. [14] also showed that aggregating the independent variable using an areal unit shape related to its spatial structure reduces the effect of MAUP, but their conclusion rely on simulated data only. To aggregate empirical data, choosing a priori areal unit shapes that preserve the spatial structure and reduce the MAUP may be challenging, and in the context of data disaggregation, may be impossible. But, in the context of data disaggregation, the MAUP bias may be smaller if the spatial units are able to capture the spatial variability of the phenomenon at hand. Recently Tuson et al. [51] proposed a theorethical and statistical framework to address the MAUP trying to detect a minimal geographical unit of analysis. Though promising, in our case the minimal geographical unit of analysis is determined by the minimal administrative level available, making the results dependent on the units used.

## MAUP scale effect

Qualitatively, fine resolution polygon training data produced predictions with a more detailed spatial pattern compared to coarser resolution training data. As far as the effect of scale on the internal precision of the model is concerned, better model precision and accuracy was reached by models trained with coarser resolution input data, contrary to what Van Boeckel et al. [21] found. These apparently contradictory results can be explained considering that Van Boeckel et al. [21] used different modelling approaches and that their goodness-of-fit were computed under a different rationale. In particular, whilst our goodness-of-fit metrics were computed between validation PSUs and predicted pixel values aggregated at the respective PSUs areas, Van Boeckel et al. [21] computed goodness-of-fit metrics between validation and predicted

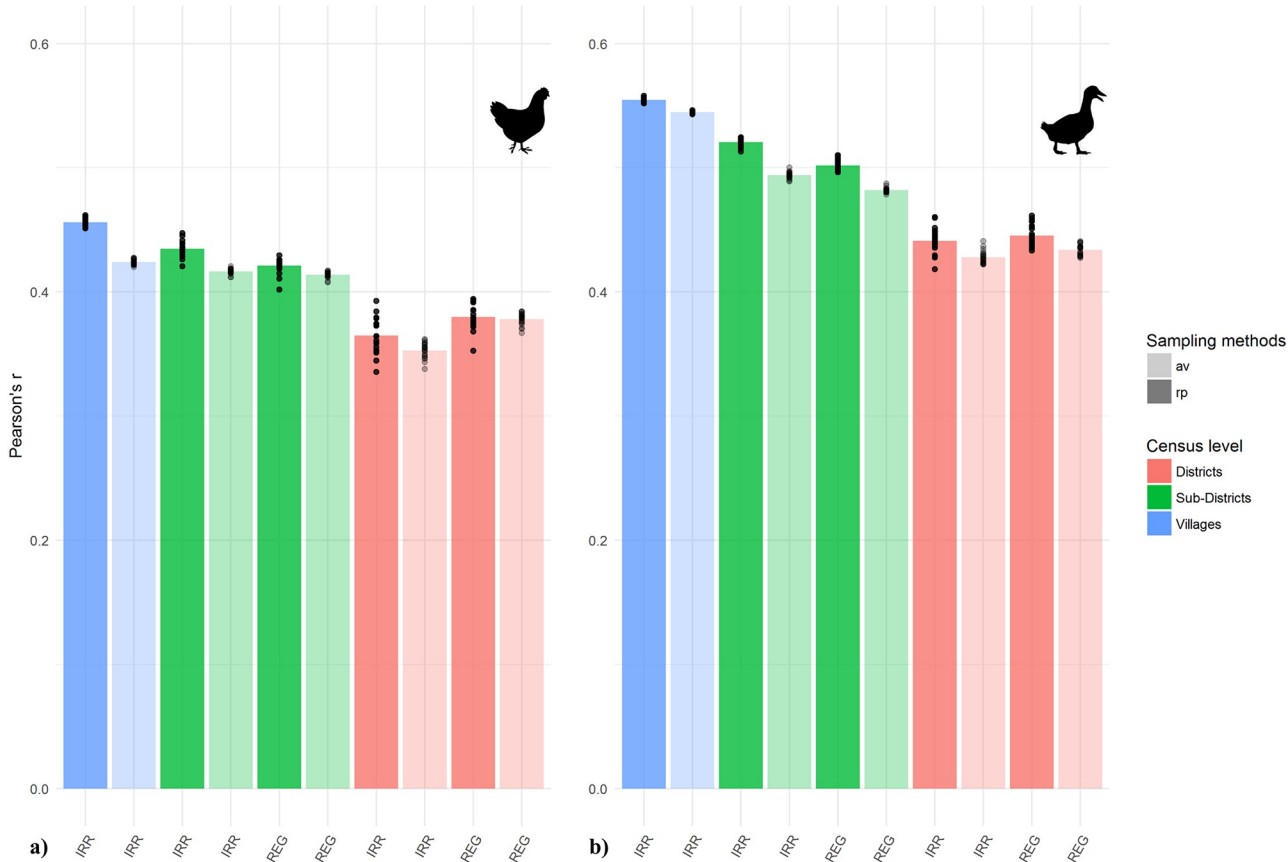

**Fig 7. Downscaling precision.** Pearson's *r* coefficient between predicted densities and observed densities at village level: a) chickens; b) ducks. Random sampling (rp), averaged sampling (av).

value at point level. Though the RMSE and COR trends are not in accordance with this previous study on Thai poultry, our results are consistent with their findings in terms of RMSE and COR ranges. More importantly, our results reflect the general trend described by Gehlke and Biehl [6], where correlation coefficients tend to increase as the number of areal units representing the data decreased, as a consequence of the data smoothing associated with the aggregation process.

## MAUP zone effect

Comparing COR and RMSE results at the same scale, REG PSU produced slightly higher mean values and less variability between model runs than IRR ones (S1 Table). In our case Pearson's *r* is the statistic most affected by the zone effect, but still it appeared marginal in comparison to the effect of scale, as observed also by Swift et al. [14] for simulated data. Recently, García-Llamas et al. [18] investigated the effects of MAUP using landscape heterogeneity as a proxy of species richness. They highlighted how the use of irregularly shaped eco-geographic area units (watersheds) performed better than arbitrary square units, probably because in their case eco-geographic areas better capture the spatial variability of species diversity. Though our REG PSU based on the ASR of the IRR PGU showed higher precision scores, our design remains affected by ecological fallacy, as both administrative levels and PSU shapes may be independent of the phenomena investigated and not effectively describe

the environmental and social envelope of farm distribution in geographical space [11, 52]. On this point, Fox et al. [52] suggest that combining reasonable assumptions to empirical data and spatial analysis may help to develop functional boundaries around the individual level investigated.

## Sampling methods effect

The choice of the sampling methods of the predictors did not affect RMSE and both correlation coefficients. The mean value of our evaluation indices were stable and the variability observable in Figs 5, 6 and 7 is likely to be more related to variability between model runs rather than to the choice of the sampling methods.

## Downscaling precision

The downscaling precision statistic was affected mainly by the scale rather than by the zone or sampling methods. The ranges are generally narrow, considering scale, zone and sampling effect. The downscaling precision as expressed by $COR_{down}$ increases with higher resolution of the training data. In fact, Robinson et al. [25] computed the goodness-of-fit metrics of their downscaling models comparing the predicted values to the observed data at the highest administrative level (a similar approach to what we used here for the $COR_{down}$), using a real census livestock dataset as we did. Similarly to our findings, they underlined how the statistical model trained on smaller administrative units got better accuracy and precision in the disaggregation of administrative units. These findings suggest that, if possible, data should be collected at the finest spatial resolution available to train the model.

The question of how to select the spatial scale of the prediction according to the available detail of aggregated data remains. The choice of the spatial scale of analysis influences the understanding of the geographical patterns [53]. When downscaling, it is thus crucial to understand whether the polygons' area within a given administrative level could influence the disaggregated results. For instance, considering the frequency histogram of district areas (S1 Fig), we do not know how larger polygons affect the downscaling precision. From one perspective, adding larger polygons would include more environmental heterogeneity in the model and would allow the model to discriminate better between suitable and unsuitable areas. However, since smaller polygons suit best in terms of downscaling precision, larger polygons could add noise to the spatial distribution of the response variable. It is unlikely that geometry of one set of areal units would match any measured phenomena exactly as it is and as it would occur for a simulated pattern [14], but new approaches combining geostatistics and Bayesian hierarchical models (e.g. [54–56]) are promising tools to address the MAUP effects.

## Conclusion

Within the GLW framework, we assessed the MAUP effects on the downscaled predictions starting from different aggregated response variable scales. We focused on the predictive rather than the explanatory power of the model, unlike numerous studies on MAUP focused on its effects on parameter estimates or p-values (e.g. [57–60]). The goal of the downscaling methodologies is not only to compare and interpret the pixel-wise absolute value per se, but also to detect and represent well the spatial variation and pattern of the phenomena investigated. Since absolute values and trends are different, the choice of $COR_{down}$ was made under the rationale to look for the scale that best preserves the observed value, allowing at the same time to detect the existing spatial trends.

GLW is an efficient approach to disaggregate census data to predict spatial distribution of livestock. Scale, rather than shapes and sampling methods, appears to affect downscaling

precision, suggesting that the finest administrative level should be sought to train the model. Moreover, the effects of MAUP appear weaker on a spatially constrained dataset rather than a more spatially homogenous one, as already shown for simulated data.

Carrying a sensitivity analysis and reporting the various results obtained from different sets of aggregation and zoning systems helped to adequately address the MAUP issue and to understand how much it affected the predictions. Understanding the magnitude of the bias introduced in the data due to the aggregation is crucial to inform spatial scientist on the often-ignored effect of data aggregation and to provide robust spatial prediction to policy maker. The effect of MAUP on aggregated data is unavoidable and only individual level data can avoid it [14, 61].

As already stated by previous authors(e.g. [14, 50, 62], sensitivity to aggregation should be analysed in any spatial study in order to correctly interpret complex results and disseminate clear and robust maps.

## Supporting information

**S1 Fig. Polygon sampling units areas' histograms.** The histograms of the area of polygon sampling units used to estimate RMSE and COR for different polygon areal sizes. The red bars represent the Average Spatial Resolution (ASR) of the polygons, while the blue lines are the polygon area classes chosen: a) 0-500 $km^2$, 500-1000 $km^2$ and >1000 $km^2$ are the districts area classes used, ASR = 3.11 km, b) 0-100 $km^2$, 100-200 $km^2$ and >200 $km^2$ are the sub-districts area classes used, ASR = 8.33 km, c) 0-10 $km^2$, 10-20 $km^2$ and >20 $km^2$ are the villages area classes used, ASR = 23.60 km.
(TIF)

**S2 Fig. Observed and predicted Log10 chicken values histogram inside *bbox*1.** The blue lines represent the mean value.
(TIF)

**S3 Fig. Observed and predicted Log10 duck values histogram inside *bbox* 1.** The blue lines represent the mean value.
(TIF)

**S4 Fig. Observed and predicted Log10 poultry values inside *bbox* 2.** a) chickens, b) Ducks.
(TIF)

**S5 Fig. Observed and predicted Log10 chickens values histogram inside *bbox* 2.** The blue lines represent the mean value.
(TIF)

**S6 Fig. Observed and predicted Log10 ducks values histogram inside *bbox* 2.** The blue lines represent the mean value.
(TIF)

**S7 Fig. Observed and predicted Log10 poultry values inside *bbox* 3.** a) chickens, b) Ducks.
(TIF)

**S8 Fig. Observed and predicted Log10 chickens values histogram inside *bbox* 3.** The blue lines represent the mean value.
(TIF)

**S9 Fig. Observed and predicted Log10 ducks values histogram inside *bbox* 3.** The blue lines represent the mean value.
(TIF)

**S1 Table. Summary table of models' goodness of fit and downscaling precision.**
(CSV)

## Acknowledgments

We thank the staff of Thailand's Department of Livestock Development (DLD), composed of the District Livestock Offices, Provincial Livestock Offices, and Center for Information Technology for animal census data; Thailand's Ministry of Transportation for geodata; and the Department of Provincial Administration, Ministry of Interior, for population data.

Computational resources have been provided by the supercomputing facilities of the Université catholique de Louvain (CISM/UCL) and the Consortium des Equipements de Calcul Intensif en Fédération Wallonie Bruxelles (CECI) funded by the Fond de la Recherche Scientifique de Belgique (FRS-FNRS).

DDR is F.R.S-FNRS Research Fellow, Belgium. DDR was supported by the FRFS-WISD Walloon Institute for Sustainable Development PDR "Mapping livestock's transition" (PDR-WISD X302317F).

## Author Contributions

**Conceptualization:** Daniele Da Re, Marius Gilbert, Pierre Bourguignon, Sophie O. Vanwambeke.

**Data curation:** Daniele Da Re, Pierre Bourguignon.

**Formal analysis:** Daniele Da Re.

**Funding acquisition:** Marius Gilbert, Sophie O. Vanwambeke.

**Methodology:** Daniele Da Re, Marius Gilbert, Sophie O. Vanwambeke.

**Project administration:** Marius Gilbert, Sophie O. Vanwambeke.

**Software:** Daniele Da Re.

**Supervision:** Marius Gilbert, Sophie O. Vanwambeke.

**Visualization:** Marius Gilbert.

**Writing – original draft:** Daniele Da Re.

**Writing – review & editing:** Daniele Da Re, Marius Gilbert, Celia Chaiban, Pierre Bourguignon, Weerapong Thanapongtharm, Timothy P. Robinson, Sophie O. Vanwambeke.

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
