## [Decision Letter · Decision Letter 0]

18 Oct 2019

PONE-D-19-20969

Downscaling livestock census data using multivariate predictive models: sensitivity to modifiable areal unit problem

PLOS ONE

Dear Mr. Da Re,

Thank you for submitting your manuscript to PLOS ONE. After careful consideration, we feel that it has merit but does not fully meet PLOS ONE’s publication criteria as it currently stands. Therefore, we invite you to submit a revised version of the manuscript that addresses the points raised during the review process.

We would appreciate receiving your revised manuscript by Dec 02 2019 11:59PM. To enhance the reproducibility of your results, we recommend that if applicable you deposit your laboratory protocols in protocols.io, where a protocol can be assigned its own identifier (DOI) such that it can be cited independently in the future. For instructions see: http://journals.plos.org/plosone/s/submission-guidelines#loc-laboratory-protocols

We look forward to receiving your revised manuscript.

Kind regards,

Sotirios Koukoulas, Ph.D

Academic Editor

PLOS ONE

Journal Requirements:

2. We note that Figure 1-3 in your submission contain map/satellite images which may be copyrighted. All PLOS content is published under the Creative Commons Attribution License (CC BY 4.0), which means that the manuscript, images, and Supporting Information files will be freely available online, and any third party is permitted to access, download, copy, distribute, and use these materials in any way, even commercially, with proper attribution. For these reasons, we cannot publish previously copyrighted maps or satellite images created using proprietary data, such as Google software (Google Maps, Street View, and Earth). For more information, see our copyright guidelines: http://journals.plos.org/plosone/s/licenses-and-copyright.

You may seek permission from the original copyright holder of Figures 1-3 to publish the content specifically under the CC BY 4.0 license. 

If you are unable to obtain permission from the original copyright holder to publish these figures under the CC BY 4.0 license or if the copyright holder’s requirements are incompatible with the CC BY 4.0 license, please either i) remove the figure or ii) supply a replacement figure that complies with the CC BY 4.0 license. Please check copyright information on all replacement figures and update the figure caption with source information. If applicable, please specify in the figure caption text when a figure is similar but not identical to the original image and is therefore for illustrative purposes only.The following resources for replacing copyrighted map figures may be helpful:

3. Please amend the manuscript submission data (via Edit Submission) to include author Sophie O. Vanwambeke.

Reviewers' comments:

Reviewer's Responses to Questions

**Comments to the Author**

1. Is the manuscript technically sound, and do the data support the conclusions?

Reviewer #1: Partly

Reviewer #2: Yes

2. Has the statistical analysis been performed appropriately and rigorously? 

Reviewer #1: Yes

Reviewer #2: Yes

3. Have the authors made all data underlying the findings in their manuscript fully available?

Reviewer #1: Yes

Reviewer #2: Yes

4. Is the manuscript presented in an intelligible fashion and written in standard English?

Reviewer #1: Yes

Reviewer #2: Yes

5. Review Comments to the Author

Reviewer #1: This paper addresses the MAUP, an important issue in spatial analysis that is related to changing spatial scale. Although the paper is well written, I recommend the authors adding clarifications for several elements as follows:

1. Statement in lines 29-31 seems not telling the complete review on the research of MAUP. There are many published studies on the MAUP issues for downscaling or spatial disaggregation.

2. Explain what the individual level data at line 77-78 are? How many of them? For the whole country or not? In which year? Etc., a detailed description of the data at individua level is required.

3. Line 76: which aggregation method was used, linear or non-linear aggregation?

4. Line 90-99: explain why those predictors were used with reference.

5. Line 97: explain why log-transform was applied to poultry density? What correction was used as mentioned in line 97 (poultry densities corrected by area).

6. Provide a map that shows the three administration units in Thailand: village, sub-district and district, how large are they relatively to each other?

7. Also provide the maps visualizing IRR and REG PSUs. Are the IRR PSUs the same as the three administration units above?

8. What are the census polygons in line 118?

9. Fig 2 has poor quality, I cannot read them all.

10. Why are there two separate sections for model evaluation and downscaling precision? Are they not the same? What is the difference between COR in line 179 and CORdown in line 187?

11. Overall, I find the findings of this paper obvious. Should this be again published the facts that we all know?

Reviewer #2: My review has turned out to be more of a proof reading session than anything else I have up;loaded an annotated pdf with quite a few very minor linguistic changes highlighted.

The science seems clean and well presented , and pretty much acceptable as it is. I have only one minor technical question - namely about the use of Voronoi polygons around villages - doesn't this mean that any density calculated for the polygon is affected by the distance between villages. I dont think this affects the validity of the study, but might it affect the values of the means calculated and this the comparison between different scales??. Not a major problem though, so I am happy to recommend acceptance with very minor revisions

6. PLOS authors have the option to publish the peer review history of their article (what does this mean?). If published, this will include your full peer review and any attached files.

Reviewer #1: No

Reviewer #2: No

---

## [Author Response · Author response to Decision Letter 0]

10 Dec 2019

Reviewers’ Comments and Authors Response

Reviewer 1

Reviewer #1: This paper addresses the MAUP, an important issue in spatial analysis that is related to changing spatial scale. Although the paper is well written, I recommend the authors adding clarifications for several elements as follows:

1. Statement in lines 29-31 seems not telling the complete review on the research of MAUP. There are many published studies on the MAUP issues for downscaling or spatial disaggregation.

A (Authors’ response): We have edited the sentence highlighting the fact that, as far as we know, the MAUP issue was more addressed in the aggregation and in relation to the changes of models estimates rather than investigating the effect of MAUP in the dowscaling precision. 

2. Explain what the individual level data at line 77-78 are? How many of them? For the whole country or not? In which year? Etc., a detailed description of the data at individua level is required.

A: We specified at L78 that the individual level data are the poultry census described in “Poultry population data” section.

3. Line 76: which aggregation method was used, linear or non-linear aggregation?

A: We used simple additive aggregation method, we have now specified it at L78 of the revised manuscript.

4. Line 90-99: explain why those predictors were used with reference.

A: We have specified it at L93-94.

5. Line 97: explain why log-transform was applied to poultry density? What correction was used as mentioned in line 97 (poultry densities corrected by area).

A: For the sake of brevity, and given the general use of these setting in the GLW framework, we refer the interested reader to the paper by Gilbert et al 2018 were this is fully explained.

6. Provide a map that shows the three administration units in Thailand: village, sub-district and district, how large are they relatively to each other?

A: We have not provided a map showing the three administrative units because at country scale it would not have been possible to appreciate the differences among them. However, we have specified the Average Spatial Resolution (ASR) of the three administrative units at L114-115.

7. Also provide the maps visualizing IRR and REG PSUs. Are the IRR PSUs the same as the three administration units above?

A: IRR and REG PSUs can be observed in Fig3. The IRR PSUs do correspond to the three administration units described above.

8. What are the census polygons in line 118?

A: The polygons used to train the model. We have called them polygons now. 

9. Fig 2 has poor quality, I cannot read them all.

A: We have improved the resolution of Fig2.

10. Why are there two separate sections for model evaluation and downscaling precision? Are they not the same? What is the difference between COR in line 179 and CORdown in line 187?

A: As specified at L42, CORdown “(...) Pearson’s r was computed between predictions and the observed data at the village level only to assess the capacity of models trained using various PSUs to predict poultry population at a fine scale”.

11. Overall, I find the findings of this paper obvious. Should this be again published the facts that we all know?

A: Though obvious, we believe that our sensitivity analysis and findings are needed to support the increase use of the GLW and other downscaled data products, in order to make the users aware of the potential issue of this data product itself and of all the downscaled data product that are becoming more and more available every year. 

Reviewer 2

12. My review has turned out to be more of a proof reading session than anything else I have up;loaded an annotated pdf with quite a few very minor linguistic changes highlighted.

A: We thank Reviewer2 for his/her positive comment. All the comments in the PDF have been addressed.

13. The science seems clean and well presented , and pretty much acceptable as it is. I have only one minor technical question - namely about the use of Voronoi polygons around villages - doesn't this mean that any density calculated for the polygon is affected by the distance between villages. I dont think this affects the validity of the study, but might it affect the values of the means calculated and this the comparison between different scales??.

In addiction Comment at L78: Doest using Voronoi polygon then mean the the animals are spread throughout the polygon...i.e the training density depends on the size of the polygons - or the distance between villages? Might it have been better to use a fixed buffer or grid instead so the poultry density wasnt affected by village density??

A: We compute the Voronoi polygons because villages administrative units for Thailand are not available. This was a simple methodological choice in order to get small scale polygons and have the chance to train the GLW model using different polygons scales. 

14. Not a major problem though, so I am happy to recommend acceptance with very minor revisions

A: We thank Reviewer2 for his/her positive assessment.

15. Comment on L218: ??coarse predicitions despite the fact that the predictors were the same resolution (500m) for all models

A: We are referring to the scale of the training polygons dataset and not to the scale and spatial resolution of the predictors. We have specified in the revised version that we are referring to the scale of the training polygons. 

16. Comment on L218: Accordingly:In agreement with ??BUT SEE your line 220 re MAUP scale effect- coarser training data gave better accuracy. Is this not a contradiction?

or have i missed the point??

A: The dowscaling precision (CORdown) measures the “(...) Pearson’s r (…) between predictions and the observed data at the village level only to assess the capacity of models trained using various PSUs to predict poultry population at a fine scale”. It is a measure of the performances of the GLW model to downscale a polygon, which is better when the training polygon is small. 

On the contrary, if we want to asses the precision of the model when predicting the whole response variable distribution, it provides better estimates when the model is trained with coarser polygons, due to the narrowing effect of the data distribution caused by the MAUP-scale effect (please see e.g. Supplementary Materials 2.1).

---

## [Editor Report · Decision Letter 1]

19 Dec 2019

Downscaling livestock census data using multivariate predictive models: sensitivity to modifiable areal unit problem

PONE-D-19-20969R1

Dear Dr. Da Re,

We are pleased to inform you that your manuscript has been judged scientifically suitable for publication and will be formally accepted for publication once it complies with all outstanding technical requirements.

With kind regards,

Sotirios Koukoulas, Ph.D

Academic Editor

PLOS ONE
---

## [Editor Report · Acceptance letter]

23 Dec 2019

PONE-D-19-20969R1 

Downscaling livestock census data using multivariate predictive models: sensitivity to modifiable areal unit problem 

Dear Dr. Da Re:

I am pleased to inform you that your manuscript has been deemed suitable for publication in PLOS ONE. Congratulations! Your manuscript is now with our production department. 

With kind regards,

on behalf of

Dr. Sotirios Koukoulas 

Academic Editor

PLOS ONE